## [Peer Review File · Nature Communications]

Editorial Note: This manuscript has been previously reviewed at another journal that is not operating a transparent peer review scheme. This document only contains reviewer comments and rebuttal letters for versions considered at Nature Communications. Mentions of the other journal and prior referee reports have been redacted.

REVIEWER COMMENTS

Reviewer #1 (Remarks to the Author):

Based on my original [redacted] review, the authors have provided a detailed response and included some of these responses as changes to the manuscript. My goal, as someone who is expert in this field, is to offer what I thought was a series of points for the authors to address to make the manuscript intelligible and accurate enough for a general scientific audience. The authors have addressed some comments and pushed back on others. At this stage, I fear the paper will still be unintelligible to much of the journal readership.

Some of my comments (such as about UC-STED) were based on trying to decipher the original manuscript, which omitted a good deal of critical experimental details (and I'd note the other reviewer shared some of my confusion). My sense is the manuscript is improved but still needs to be cleaned up for most readers of Nature Communications. And, frankly, there are still errors. A few key things :

1) Calling these "barcodes": the goal of barcodes is to identify unique components of a complex mixture. Barcodes are used to quickly identify individual components from mixtures of thousands or millions of components. I do not think barcodes and multiplexed imaging are the same; barcodes are used for sorting or selecting, whereas "multiplexing" is simply a fancy (and, I think, unnecessary) word for multicolor imaging. I think what the authors are proposing with their structures is high-resolution information storage using small libraries of nanorods.

To me, being able to distinguish distinct nanorods mixed together on a slide is an interesting characterization of those nanorods; but barcoding requires the added step of linking known barcodes to individual components of a mixture for sorting or selecting, and is an entirely different experiment. As it stands, I think most readers interesting in barcoding will be terribly confused by how these structures might be applied. I simply think "barcode" is the wrong word for this study.

2) The authors suggest 15^4 nanorods are possible (that equation need to be solved in the text), but of course this would require that many unique tagging methods for most or all of the suggested applications. Specifically, this sentence strikes me as an exceptionally ambitious application requiring much new specific and orthogonal nanoparticle bioconjugation chemistry to be developed: "The developed geometrical nanobarcode could be simultaneously used for tagging different cellular species and identified by the super-resolution nanoscopy, bypassing the need for the stepwise addition of dyes for multiplexed imaging of intracellular structures.^{47,48}" I would also wonder how 50000 nanorods can be described as realistic, for a complex multisegmented nanorod synthesis. Mix and split may be possible, but not at that scale for inorganic reactions.

I think the paragraph on these suggested applications needs to be scaled back considerably.

3) Calling these "one-dimensional": the correct nanoscience term for these is "nanorods". 1D compounds (carbon nanotubes, Si nanowires, etc) have a dimension less than 10 nm, and usually closer to 1 nm. Just because something is shorter on one axis than the other 2 does not make it 1-dimensional.

4) This new claim of primacy: "For the first time, we precisely localize the position of closely packed sensitizer sections and visualize the energy transfer effect at nanoscale" does not hold up. There have been many core/shell and core/shell/shell UCNPs in which sensitizers and activators have been segregated to different domains of the nanostructure, sometimes with domains added to facilitate energy transfer (eg X Liu, using Gd to transfer energy between domains; or many previous Nd/Yb/Er constructs; or Yb "active shells" with Yb/Er cores; or dye-sensitized Yb/Er UCNPs, etc).

4) Much of the language is still overinflated and unclear. The word "super" is used a lot which, outside of "super resolution", is probably a mistake.

Reviewer #2 (Remarks to the Author):

[redacted]

Additional Comments on re-review: The authors have made significant additions to the paper, including extensive supplemental materials to clarify the Emission Saturation Microscopy imaging method they are using, as reported in their 2018 Nature Comm. paper[1].

It would be good to point out that the Abbe limit is a mathematical calculation referring to the distance between two Airy functions below which practical measurements distinguishing them become difficult based on an agreed upon criterion. Resolution is essentially a signal to noise issue, and one sees in the images in the paper that this is an issue for the data shown. In examining the images, it is clear that $\approx 100\text{nm}$ resolution is clearly demonstrated, but that the images with finer features become more difficult, even after a deconvolution (I am assuming the measured PSF is used in this process, but it's not clear to me).

Another other point is the saturation effect is a non-linear effect and this makes deconvolution more tricky, we have to assume that the nonlinearity is the same in all instances, or it would change the shape of the PSF and it would no longer be translationally invariant. A sophisticated deconvolution algorithm could be conceived, but it would most certainly rely on sparse objects. This does not bode well for objects closely spaced and likely is affecting the data presented.

The last point is about the independence of the optical signatures, or so-called dimensions, the emission wavelength, excitation wavelength, intensity dependence and lifetime. These are actually interdependent so it's not clear to me how easily or reliable they would be to uniquely identify the barcode in question. It's easy to imagine scenarios where these could get mixed up in the reading of two dissimilar barcodes or two closely spaced barcodes. This seems not to be discussed in the paper. I see only emission intensity images but didn't find representations of these other dimensions. Figure 5 shows one example of nominally identical barcodes. It's not clear, but the assumption is the different segments display different properties, are they all the same but different properties are visualized for each segment? It's not a clear demonstration of encoding.

Originality and significance:

[redacted]

[redacted]

Additional comments on re-review: The authors have clarified the mechanisms for superresolution in this draft. A demonstration of two nominally identical barcodes placed less than 100nm apart is shown in the modified figure 2. The image is not particularly clear and appears to have been significantly processed. This is an issue in general with the method being used, as it relies on a decrease in photon counts which necessarily brings with it a decrease in signal to noise. This is apparent in much of the data shown throughout the paper and supplemental materials.

Data & methodology:[redacted]

Additional comments on re-review: I think the discussion around the mechanisms for superresolution are clearer and improved.

Appropriate use of statistics and treatment of uncertainties: [redacted]

Additional comments on re-review: No additional comments.

Conclusions, robustness, validity, reliability: [redacted]

Additional comments on re-review: The papers conclusions are clearer than the previous submission.

Suggested improvements: [redacted]

Additional comments on re-review: My opinion regarding the usefulness of the barcodes is essentially unchanged. I do think there will be issues deploying these in high density applications. The English language usage in the paper is improved, but could still use editing.

References, clarity and context: Please see above comments.

- (1) Chen, C.; Wang, F.; Wen, S.; Su, Q. P.; Wu, M. C. L.; Liu, Y.; Wang, B.; Li, D.; Shan, X.; Kianinia, M.; et al. Multi-Photon near-Infrared Emission Saturation Nanoscopy Using Upconversion Nanoparticles. *Nat. Commun.* 2018, 9, 4–9.
- (2) Zhang, Y.; Zhang, L.; Deng, R.; Tian, J.; Zong, Y.; Jin, D.; Liu, X. Multicolor Barcoding in a Single Upconversion Crystal. *J. Am. Chem. Soc.* 2014, 136, 4893–4896.
- (3) Liu, D.; Xu, X.; Du, Y.; Qin, X.; Zhang, Y.; Ma, C.; Wen, S.; Ren, W.; Goldys, E. M.; Piper, J. A.; et al. Three-Dimensional Controlled Growth of Monodisperse Sub-50 Nm Heterogeneous Nanocrystals. *Nat. Commun.* 2016, 7, 1–8.
- (4) Liu, Y.; Lu, Y.; Yang, X.; Zheng, X.; Wen, S.; Wang, F.; Vidal, X.; Zhao, J.; Liu, D.; Zhou, Z.; et al. Amplified Stimulated Emission in Upconversion Nanoparticles for Super-Resolution Nanoscopy. *Nature* 2017.

Reviewer #1 (Remarks to the Author):

Based on my original [redacted] review, the authors have provided a detailed response and included some of these responses as changes to the manuscript. My goal, as someone who is expert in this field, is to offer what I thought was a series of points for the authors to address to make the manuscript intelligible and accurate enough for a general scientific audience. The authors have addressed some comments and pushed back on others. At this stage, I fear the paper will still be unintelligible to much of the journal readership.

Some of my comments (such as about UC-STED) were based on trying to decipher the original manuscript, which omitted a good deal of critical experimental details (and I'd note the other reviewer shared some of my confusion). My sense is the manuscript is improved but still needs to be cleaned up for most readers of Nature Communications. And, frankly, there are still errors. A few key things:

Comment 1: Calling these "barcodes": the goal of barcodes is to identify unique components of a complex mixture. Barcodes are used to quickly identify individual components from mixtures of thousands or millions of components. I do not think barcodes and multiplexed imaging are the same; barcodes are used for sorting or selecting, whereas "multiplexing" is simply a fancy (and, I think, unnecessary) word for multicolor imaging. I think what the authors are proposing with their structures is high-resolution information storage using small libraries of nanorods.

To me, being able to distinguish distinct nanorods mixed together on a slide is an interesting characterization of those nanorods; but barcoding requires the added step of linking known barcodes to individual components of a mixture for sorting or selecting, and is an entirely different experiment. As it stands, I think most readers interesting in barcoding will be terribly confused by how these structures might be applied. I simply think "barcode" is the wrong word for this study.

Response: To avoid misleading the readers, we only use the nanorods in our manuscript. We deleted Figure 5 and related claims, and changed the title as "Nanorods with multidimensional optical information beyond diffraction limit".

Comment 2: The authors suggest 15^4 nanorods are possible (that equation need to be solved in the text),

but of course this would require that many unique tagging methods for most or all of the suggested applications. Specifically, this sentence strikes me as an exceptionally ambitious application requiring much new specific and orthogonal nanoparticle bioconjugation chemistry to be developed: “The developed geometrical nanobarcodes could be simultaneously used for tagging different cellular species and identified by the super-resolution nanoscopy, bypassing the need for the stepwise addition of dyes for multiplexed imaging of intracellular structures.^{47,48}” I would also wonder how 50000 nanorods can be described as realistic, for a complex multisegmented nanorod synthesis. Mix and split may be possible, but not at that scale for inorganic reactions.

I think the paragraph on these suggested applications needs to be scaled back considerably.

Response: We agree with the reviewer and deleted claims of barcoding and suggested applications that are not fully demonstrated in this study. This would improve the focus of our current study.

Comment 3: Calling these “one-dimensional”: the correct nanoscience term for these is “nanorods”. 1D compounds (carbon nanotubes, Si nanowires, etc) have a dimension less than 10 nm, and usually closer to 1 nm. Just because something is shorter on one axis than the other 2 does not make it 1-dimensional.

Response: We deleted the “one-dimensional” and corrected the related description in our revised manuscript.

Comment 4: This new claim of primacy: “For the first time, we precisely localize the position of closely packed sensitizer sections and visualize the energy transfer effect at nanoscale” does not hold up. There have been many core/shell and core/shell/shell UCNPs in which sensitizers and activators have been segregated to different domains of the nanostructure, sometimes with domains added to facilitate energy transfer (eg X Liu, using Gd to transfer energy between domains; or many previous Nd/Yb/Er constructs; or Yb “active shells” with Yb/Er cores; or dye-sensitized Yb/Er UCNPs, etc).

Response: We thank the reviewer to point this out. Actually, we want to claim we visualized the energy transfer effect at nanoscale for the first time. To avoid any misunderstanding, we deleted “for the first time” in our revised manuscript.

Comment 5: Much of the language is still overinflated and unclear. The word “super” is used a lot which, outside of “super resolution”, is probably a mistake.

Response: We revised our manuscript carefully to avoid any potential overinflated and unclear information. Also, we deleted the super outside of “super resolution” in our revised manuscript.

Reviewer #2 (Remarks to the Author):

Reviewer 2 has very clear suggestions for us by putting the previous comments on our original *Nature* submission as well as the additional comments here. As we have responded the previous comments in last version, here we only list the additional comments on re-review.

Summary of key results: [redacted]

Additional Comments on re-review: The authors have made significant additions to the paper, including extensive supplemental materials to clarify the Emission Saturation Microscopy imaging method they are using, as reported in their 2018 *Nature Comm.* paper[1].

Comment 1: It would be good to point out that the Abbe limit is a mathematical calculation referring to the distance between two Airy functions below which practical measurements distinguishing them become difficult based on an agreed upon criterion. Resolution is essentially a signal to noise issue, and one sees in the images in the paper that this is an issue for the data shown. In examining the images, it is clear that $\approx 100\text{nm}$ resolution is clearly demonstrated, but that the images with finer features become more difficult, even after a deconvolution (I am assuming the measured PSF is used in this process, but it's not clear to me).

Response: We agree with the reviewer that the signal to noise is another important parameter for resolving features. As low signal-to-noise ratio will affect the effective image resolution, we set the length of each functional section as 20 nm for providing a bright emission to achieve a signal-to-noise ratio higher than 5. Also, the signal-to-noise ratio can be further improved by increasing the volume and the intensity of each sections of the heterogeneous nanorods according to our recent work (*Liu et al. Small 2020, 16, 1905572*), in which we achieved the mapping of single nanoparticles located 55 μm inside a tumor spheroid with a resolution of 98 nm. To make it clear, we have added related discussion into the main text to emphasize the importance of the signal strength. See also below:

“As based on the mathematical calculation of Abbe limit the spatial resolution of our emission saturation super-resolution microscopy is 29 nm (Fig. S7 b-c), the limit in resolving the two active segments needs to be longer than the distance of 50 nm (see our simulation results in Fig. S7d-f). As the resolution is also dependent on the signal-to-noise ratio of the images, we set the length of each functional segment as 20 nm to achieve sufficiently bright with a signal-to-noise ratio larger than 5.”

“The signal-to-noise ratio can be further improved by increasing the width, the volume and the brightness of each segment of the heterogeneous nanorods.”³⁷

Comment 2: Another other point is the saturation effect is a non-linear effect and this makes deconvolution more tricky, we have to assume that the nonlinearity is the same in all instances, or it would change the shape of the PSF and it would no longer be translationally invariant. A sophisticated deconvolution algorithm could be conceived, but it would most certainly rely on sparse objects. This does not bode well for objects closely spaced and likely is affecting the data presented.

Response: We thank the reviewer for pointing this out. We understand the reviewer’s concern that the nonlinear response will generate a non-standard doughnut pattern, which affects the deconvolution. We are aware of this, and in our deconvolution process, we did not use any pattern function to fit our beam; hence, the non-linear effect does not affect the deconvolution process.

Specifically, we used the subtraction algorithm to deconvolute our images according to a previous report (Kuang et al. *Sci Rep* **3**, 1441, 2013, reference #7 in SI). In this process, two types of scanning images are required. One is the “negative-contrast image” (doughnut PSF) captured by scanning the sample by doughnut beam as shown in Figure S9b. The other is the “confocal image” (Gaussian PSF) generated by either scanning the sample by Gaussian beam or applying a low pass image filter on the “negative-contrast image” (Figure S9a). The final positive-contrast super-resolution image has been constructed by the intensity subtraction of these two images as shown in Figure S9c. We can use equation 1 to illustrate the subtraction processing.

$$I_{pos} = I_{con} - r \times I_{neg} \quad (1)$$

Where I_{con} , I_{neg} , and I_{pos} are the normalized intensity distribution of the confocal, negative-contrast super-resolution image, and positive-contrast super-resolution image, respectively. r is the subtractive factor used to avoid the negative values to improve the image quality.

Figure S9. Illustration of the positive-contrast super-resolution imaging process. (a) Confocal image of upconversion nanorods. (b) Negative-contrast super-resolution image of upconversion nanorods. (c) The processed positive-contrast super-resolution image of upconversion nanorods. Dwell time is 2ms. Scale bar is 500 nm.

To make it clear, we added the above information in the revised supporting information, as below:

“Here we use a subtraction algorithm to obtain the deconvoluted super-resolution images.⁷ Figure S9 shows the key process based on the two images, i.e. the “negative-contrast image” captured by scanning the sample by doughnut beam (Figure S9b), and the “confocal image” generated by either scanning the sample by Gaussian beam or applying a low pass image filter on the “negative-contrast image” (Figure S9a). The final positive-contrast super-resolution image (Figure S9c) is constructed by the intensity subtraction of the two images. We use equation 1 to illustrate the subtraction process.

$$I_{pos} = I_{con} - r \times I_{neg} \quad (1)$$

Where I_{con} , I_{neg} , and I_{pos} are the normalized intensity distributions of the confocal, negative-contrast super-resolution image, and positive-contrast image, respectively. r is the subtractive factor used to avoid the negative values to improve the image quality.”

Comment 3: The last point is about the independence of the optical signatures, or so-called dimensions, the emission wavelength, excitation wavelength, intensity dependence and lifetime. These are actually interdependent so it's not clear to me how easily or reliable they would be to uniquely identify the barcode in question. It's easy to imagine scenarios where these could get mixed up in the reading of two dissimilar barcodes or two closely spaced barcodes. This seems not to be discussed in the paper. I see only emission intensity images but didn't find representations of these other dimensions. Figure 5 shows one example of nominally identical barcodes. It's not clear, but the assumption is the different segments display different properties, are they all the same but different properties are visualized for each segment? It's not a clear demonstration of encoding.

Response: There are four optical dimensions employed in this work, including emission wavelength, excitation wavelength, lifetime and excitation power dependence. Only lifetime and excitation power dependence have the similar features as shown in figure 3a,b and figure S14. There is no interdependence for emission wavelength, excitation wavelength and excitation power dependence. Emission wavelength depends on the type of activators as shown in figure S12, where we use the 650 nm emission of Er and 800 nm emission of Tm as an example in this study. Excitation wavelength depends on the type of sensitizers as show in figure 3e-h, where we use the 980 nm excitation of Yb and 800 nm excitation of Nd as an example in this study. Lifetime and excitation power dependence is depend on the concentrations of the activators as shown in Figure 3a-d, where we use 4% and 10% Tm as an example in this study. So it is easily and reliable to uniquely identify the barcode in question.

Yes, Figure 5 shows one example of multidimensional barcodes with different segments display different properties. These different segments have different dopant combination. As show in previous Figure S17, there are four different segments doped with 6% Er, 60% Nd, 10%Tm and 4%Tm, respectively, which have the emission at different optical dimensions.

Originality and significance: [redacted]

[redacted]

These are previous comments.

Comment 4: Additional comments on re-review: The authors have clarified the mechanisms for superresolution in this draft. A demonstration of two nominally identical barcodes placed less than 100nm apart is shown in the modified figure 2. The image is not particularly clear and appears to have been significantly processed. This is an issue in general with the method being used, as it relies on a decrease in photon counts which necessarily brings with it a decrease in signal to noise. This is apparent in much of the data shown throughout the paper and supplemental materials.

Response: We appreciate the reviewer's summary on our revised manuscript. We agree with the reviewer that the signal-to-noise ratio is important for the high quality super-resolution imaging. The signal-to-noise ratio can be further improved by increasing the width, the volume and the brightness of each section of the heterogeneous nanorods.

Data & methodology: [redacted]

This is previous comment.

Additional comments on re-review: I think the discussion around the mechanisms for superresolution are clearer and improved.

We thank the reviewer for his/her support.

10. Appropriate use of statistics and treatment of uncertainties: [redacted]

This is previous comment.

11. Additional comments on re-review: No additional comments.

12. Conclusions, robustness, validity, reliability: [redacted]

This is previous comment.

13. Additional comments on re-review: The papers conclusions are clearer than the previous submission.

We thank the reviewer for his/her support.

14. Suggested improvements: [redacted]

This is previous comment.

15. Additional comments on re-review: My opinion regarding the usefulness of the barcodes is essentially unchanged. I do think there will be issues deploying these in high density applications. The English language usage in the paper is improved, but could still use editing.

Response: We agree with the reviewer that these barcodes are not easy for use in high-density bio-applications and deleted the related claims. We have also improved our English language in our revised manuscript.

16. References, clarity and context: Please see above comments.

(1) Chen, C.; Wang, F.; Wen, S.; Su, Q. P.; Wu, M. C. L.; Liu, Y.; Wang, B.; Li, D.; Shan, X.; Kianinia, M.; et al. Multi-Photon near-Infrared Emission Saturation Nanoscopy Using Upconversion Nanoparticles. *Nat. Commun.* 2018, 9, 4–9.

(2) Zhang, Y.; Zhang, L.; Deng, R.; Tian, J.; Zong, Y.; Jin, D.; Liu, X. Multicolor Barcoding in a Single Upconversion Crystal. *J. Am. Chem. Soc.* 2014, 136, 4893–4896.

(3) Liu, D.; Xu, X.; Du, Y.; Qin, X.; Zhang, Y.; Ma, C.; Wen, S.; Ren, W.; Goldys, E. M.; Piper, J. A.; et al. Three-Dimensional Controlled Growth of Monodisperse Sub-50 Nm Heterogeneous Nanocrystals. *Nat. Commun.* 2016, 7, 1–8.

(4) Liu, Y.; Lu, Y.; Yang, X.; Zheng, X.; Wen, S.; Wang, F.; Vidal, X.; Zhao, J.; Liu, D.; Zhou, Z.; et al. Amplified Stimulated Emission in Upconversion Nanoparticles for Super-Resolution Nanoscopy. *Nature* 2017.

REVIEWER COMMENTS

Reviewer #1 (Remarks to the Author):

My concerns have been addressed, and I appreciate the authors made these changes.

Note, the current title (and some things in the manuscript) need to be edited for correct grammar. "Nanorods with multidimensional optical information beyond diffraction limit" needs a "the" before diffraction.

Reviewer #2 (Remarks to the Author):

Summary of key results :

Additional comments on tertiary review :

In summary, the authors provide a refined synthetic approach[1,2] to producing stratified layers is a rare-earth doped host with bi-lateral symmetry and use a Laguerre-Gaussian beam to image them using the nonlinear imaging method known as emission saturation nanoscopy reported earlier by the authors in 2018[3]. The new manuscript title, "Nanorods with multidimensional optical information beyond diffraction limit," has culled bar-coding terminology from the paper but retained the notion of subwavelength information encoded in a nanostructured rare-earth doped nanorod and super-resolution decoding.

Originality and significance :

Additional comments on tertiary review :

The authors addressed comments about signal to noise ratio and its relation to resolution by adding some discussion in this regard. After examining figure 2 again, it seems to me the resolution can be no better than about 100nm as seen in the series of images in figure 2d, and may in general be lower. These images are essentially images of the point spread function of the donut-mode excitation and the bilateral punctate structure of the nanorods. While it is true that the nonlinearity of the upconversion luminescence will narrow the PSF, if the nanorods are closer than approximately the 100nm distance to each other, they would obviously be indistinguishable and any information they encode would become muddled. A proper deconvolution would replace each punctated bi-donut image by a Gaussian spot, accounting for the angular dependence (non-cylindrical symmetry). If two of these Gaussian spots were sufficiently distinct, they would be resolvable according to an agreed upon criteria, e.g. Rayleigh's criteria. The structure of the point spread function is not the limiting factor here, it is the nanorod structure itself. Even if we have a method to localize these bi-punctate spots individually, there is no obvious mechanism to distinguish one from the other. Consider there are an infinite number of orientations corresponding to the rotation of these nanorods about their axis of symmetry. We could not retrieve this structure in a dense labeling scenario – it presents only further complications in interpreting the imaging data. Resolution and localization are not the same thing. In this paper, we are imaging the convolution of the cylindrical PSF and the nanorod oriented at a given angle, allowed in these images due to the fact that the bars are sparsely distributed so there can be no ambiguity. The resolution called out in figure 2f is predicated on our knowledge of the nanorod geometry (the rod visage is even superimposed on the bi-punctate image), this necessitates the nanorods are not overlapping and well separated. This requirement is part of the localization procedure employed. Without this knowledge, we would be left to guess which spots belong to which nanorod, confounding our ability to resolve the nanorods. To avoid this, one should place the nanorods a distance greater than the bi-punctate spot separation, e.g. greater than 100nm based on signal to noise demonstrated in the images shown in the paper. The superresolution aspects of the paper remain weakly motivated.

The authors reply that the excitation intensity, emission decay rate and wavelength are independent. It is correct that the rare-earth ion multiplet energies are generally independent of the intensity, but the

emission intensity, including the spectral distribution, will depend on the populations of the levels feeding the emitting state as well as other relaxation channels in and out of that state, which will depend on excitation intensity and will affect the spectral signature as well as the emission decay rate. Because of these nonlinear couplings, the intensity, emission energy and decay rate are not independent. The information carrying capacity of the nanorods is therefore overstated.

Data & methodology:

Additional comments on tertiary review:

I believe the imaging components of this manuscript need to be reconsidered. There may be some interesting opportunities, if the bi-punctate spacing can be chosen judiciously it could affect the spatial frequencies detectable. The orientation of the nanorods may also present an opportunity in specific applications where more information than x,y coordinates are desired.

Appropriate use of statistics and treatment of uncertainties: No comment.

Additional comments on tertiary review: No additional comments.

Conclusions, robustness, validity, reliability:

Additional comments on tertiary review:

Again, I want to complement the authors on their exquisitely prepared materials, they are of high quality. I think revisiting the imaging aspects of this work is warranted.

Suggested improvements:

Additional comments on tertiary review:

As the barcode element of the paper has been (superficially) removed, the major contribution is the fabrication of these nanorods and their variety of possible manifestations. These could encode information, surely. The bilateral symmetry, combined with the donut excitation imaging, may also have a use. However, I don't believe it will be useful in most superresolution applications. These might be good for low density labeling applications and additionally where the orientation of the nanorod could give some important information. In light of the many recent papers on high quality growth and characterization of these materials[4], I think the authors should think carefully about the work and decide if the nanofabrication aspect is a significant advancement in and of itself and whether the imaging aspects may be further developed.

References, clarity and context: Please see above comments.

- (1) Zhang, Y.; Zhang, L.; Deng, R.; Tian, J.; Zong, Y.; Jin, D.; Liu, X. Multicolor Barcoding in a Single Upconversion Crystal. *J. Am. Chem. Soc.* 2014, 136, 4893–4896.
- (2) Liu, D.; Xu, X.; Du, Y.; Qin, X.; Zhang, Y.; Ma, C.; Wen, S.; Ren, W.; Goldys, E. M.; Piper, J. A.; et al. Three-Dimensional Controlled Growth of Monodisperse Sub-50 Nm Heterogeneous Nanocrystals. *Nat. Commun.* 2016, 7, 1–8.
- (3) Chen, C.; Wang, F.; Wen, S.; Su, Q. P.; Wu, M. C. L.; Liu, Y.; Wang, B.; Li, D.; Shan, X.; Kianinia, M.; et al. Multi-Photon near-Infrared Emission Saturation Nanoscopy Using Upconversion Nanoparticles. *Nat. Commun.* 2018, 9, 4–9.
- (4) Wen, S.; Zhou, J.; Zheng, K.; Bednarkiewicz, A.; Liu, X.; Jin, D. Advances in Highly Doped Upconversion Nanoparticles. *Nat. Commun.* 2018, 9, 2415.

We are appreciative of the efforts and time invested by editor and reviewers in this process. The comments and recommendations were constructive, and taking them into account has improved the clarity and scientific rigor of the manuscript. The reviewers appreciated the artful and high-quality control of nanomaterial synthesis and super-resolution characterization.

The major queries arose from the limitation for the high-density labelling and the independence of different dimensions. In our revised manuscript, we added the open discussion about the limitation of our nanorods and removed some claims about the independence of different dimensions according to the reviewer's suggestions. We also added the discussion about the advantages of the orientation information inside the nanorods according to the reviewer's very constructive suggestion.

Please find our detailed replies (in black) to the reviewers' comments (in blue).

Reviewers' Comments:

Reviewer #1 (Remarks to the Author):

My concerns have been addressed, and I appreciate the authors made these changes.

Note, the current title (and some things in the manuscript) need to be edited for correct grammar. "Nanorods with multidimensional optical information beyond diffraction limit" needs a "the" before diffraction.

Response: We thank the reviewer for his/her recommendation and suggestion. In the revised manuscript, we changed the title as "Nanorods with multidimensional optical information beyond the diffraction limit".

Reviewer #2 (Remarks to the Author):

Summary of key results:

Additional comments on tertiary review:

Comment 1. In summary, the authors provide a refined synthetic approach[1,2] to producing stratified layers is a rare-earth doped host with bi-lateral symmetry and use a Laguerre-Gaussian beam to image them using the nonlinear imaging method known as emission saturation nanoscopy reported earlier by the authors in 2018[3]. The new manuscript title, "Nanorods with multidimensional optical information beyond diffraction limit," has culled bar-coding terminology from the paper but retained the notion of subwavelength information encoded in a nanostructured rare-earth doped nanorod and super-resolution decoding.

Response: We thank the reviewer for his/her summary of our key results.

Originality and significance:

Additional comments on tertiary review:

Comment 2. The authors addressed comments about signal to noise ratio and its relation to resolution by adding some discussion in this regard. After examining figure 2 again, it seems to me the resolution can be no better than about 100nm as seen in the series of images in figure 2d, and may in general be lower. These images are essentially images of the point spread function of the donut-mode excitation and the bilateral punctate structure of the nanorods. While it is true that the nonlinearity of the upconversion luminescence will narrow the PSF, if the nanorods are closer than approximately the 100nm distance to each other, they would obviously be indistinguishable and any information they encode would become muddled. A proper deconvolution would replace each punctated bi-donut image by a Gaussian spot, accounting for the angular dependence (non-cylindrical symmetry). If two of these Gaussian spots were sufficiently distinct, they would be resolvable according to an agreed upon criteria, e.g. Rayleigh's criteria. The structure of the point spread function is not the limiting factor here, it is the nanorod structure itself. Even if we have a method to localize these bi-punctate spots individually, there is no obvious mechanism to distinguish one from the other. Consider there are an infinite number of orientations corresponding to the rotation of these nanorods about their axis of symmetry. We could not retrieve this structure in a dense labeling scenario – it presents only further complications in interpreting the imaging data. Resolution and localization are not the same thing. In this paper, we are imaging the convolution of the cylindrical PSF and the nanorod oriented at a given angle, allowed in these images due to the fact that the bars are sparsely distributed so there can be no ambiguity. The resolution called out in figure 2f is predicated on our knowledge of the nanorod geometry (the rod visage is even superimposed on the bi-punctate image), this necessitates the nanorods are not overlapping and well separated. This requirement is part of the localization procedure employed. Without this knowledge, we would be left to guess which spots belong to which nanorod, confounding our ability to resolve the nanorods. To avoid this, one should place the nanorods a distance greater than the bi-punctate spot separation, e.g. greater than 100nm based on signal to noise demonstrated in the images shown in the paper. The superresolution aspects of the paper remain weakly motivated.

Response: We thank the reviewer for pointing out this limitation. We agree with the reviewer on the limitation of our current nanorod structure itself. The samples should be well prepared with the distance of nanorods greater than the bi-punctate spot separation. To make it clear, we have added the related discussion in the main text:

“It should be noted that ideally the distance of any two nearby nanorods should be greater than the bi-punctate spot separation (100 nm for samples in figure 2f) to determine which pair of spots belong to which nanorod. This makes our nanorods suitable for low density labelling applications.”

Comment 4. The authors reply that the excitation intensity, emission decay rate and wavelength are independent. It is correct that the rare-earth ion multiplet energies are generally independent of the intensity, but the emission intensity, including the spectral distribution, will depend on the populations of the levels feeding the emitting state as well as other relaxation channels in and out of that state, which will depend on excitation intensity and will affect the spectral signature as well as the emission decay rate. Because of these nonlinear couplings, the intensity, emission energy and decay rate are not independent. The information carrying capacity of the nanorods is therefore overstated.

Response: There might be some misunderstanding about the independent dimensions. We reply “there is no interdependence for emission wavelength, excitation wavelength and excitation power dependence” in our previous response letter, instead of “excitation intensity, emission decay rate and wavelength”.

The emission wavelengths are from different kinds of activators instead of different emitting states of the same activator, so the power-dependent emitting state and spectral signature of same activator will not affect the dimension of emission wavelength. For example, the 650 nm emission of Er and the 800 nm emission of Tm are employed in this study (Figure S12) and their emission wavelength (at 650 nm and 800 nm) won't shift with excitation power and excitation wavelength. Therefore, the emission wavelengths are independent of the excitation intensity and excitation wavelength.

The excitation wavelengths are determined by the different types of sensitizers instead of different absorption energy levels of the activator, as shown in Figure 3e-h, where we used the 980 nm excitation of Yb and 800 nm excitation of Nd as an example in this study. Therefore, the excitation wavelengths are independent of the excitation intensity and emission wavelength.

For excitation intensity, selective activation of two sections doped by the relatively lower concentration of activators (4% Tm³⁺) and higher concentration of activators (10% Tm³⁺) can be achieved by using low and high power densities of 980 nm excitation, respectively (Fig. 3b). At low excitation power, the sections with high doping concentration of activators cannot be active due to serious cross-relaxation and only the sections with low-doping concentration of activators show the negative bi-punctate signals. At the high excitation power, the sections with the low-doping concentration of activators are saturated and have the same image with the confocal scan without the negative bi-punctate signals, while the section with highly-doped activators show the negative bi-punctate signals. Therefore, excitation intensity dimension is independent with the excitation wavelength and emission wavelength.

To avoid any potential overstating, we revised the related description and deleted the “independent” and “orthogonal” in the revised manuscript.

Data & methodology:

Additional comments on tertiary review:

Comment 5. I believe the imaging components of this manuscript need to be reconsidered. There may be some interesting opportunities, if the bi-punctate spacing can be chosen judiciously it could affect the spatial frequencies detectable. The orientation of the nanorods may also present an opportunity in specific applications where more information than x,y coordinates are desired.

Response: We thank the reviewer for the very constructive suggestions and pointing out these important advantages. This is indeed a great point “if the bi-punctate spacing can be chosen judiciously it could affect the spatial frequencies detectable.” We will try to use these features in our further works. Also, the orientation information of the nanorods indeed present an opportunity

in specific applications. We added the related discussion in the revised manuscript:

“The orientation information inside the bi-punctate nanorods structure presents an opportunity for future applications where the dipole orientation information in three coordinates are desired.”³⁷

Appropriate use of statistics and treatment of uncertainties: No comment.

Additional comments on tertiary review: No additional comments.

Conclusions, robustness, validity, reliability:

Additional comments on tertiary review:

Comment 6. Again, I want to complement the authors on their exquisitely prepared materials, they are of high quality. I think revisiting the imaging aspects of this work is warranted.

Response: We appreciate the reviewer’s encouragement for our materials part. We have revisited the imaging aspect according the reviewer’s comments (#2-5) to discuss the limitation in the high-density labelling and the potential advantages in the orientation imaging.

Suggested improvements:

Additional comments on tertiary review:

Comment 7. As the barcode element of the paper has been (superficially) removed, the major contribution is the fabrication of these nanorods and their variety of possible manifestations. These could encode information, surely. The bilateral symmetry, combined with the donut excitation imaging, may also have a use. However, I don’t believe it will be useful in most superresolution applications. These might be good for low density labeling applications and additionally where the orientation of the nanorod could give some important information. In light of the many recent papers on high quality growth and characterization of these materials[4], I think the authors should think carefully about the work and decide if the nanofabrication aspect is a significant advancement in and of itself and whether the imaging aspects may be further developed.

Response: We thank the reviewer for the very constructive suggestions for the future works. We agree with the reviewer that “These might be good for low density labeling applications and additionally where the orientation of the nanorod could give some important information.”

For the nanofabrication, we have the significant advancement with the anisotropic heterogeneous features compared with the many recent papers, as most of the current UCNP are sphere ones with the isotropic core@shell structures (we have summarized them in Ref 4). We anticipate the many photophysics could be discovered from the geometrically distinct nanostructures, including the dynamics and direction of energy transfer in heterogeneous nanomaterials. We have the related discussion in the revised manuscript. See also below:

“While this work provides new scopes for controlled fabrication and sub-diffraction-limit characterizations of the anisotropic heterogeneous nanomaterials, we anticipate the many photophysics to be discovered from the new library of geometrically distinct nanostructures,

*including the dynamics and directions of energy transfer process within the heterogeneous nanomaterials.*⁴⁵”

References, clarity and context: Please see above comments.

(1) Zhang, Y.; Zhang, L.; Deng, R.; Tian, J.; Zong, Y.; Jin, D.; Liu, X. Multicolor Barcoding in a Single Upconversion Crystal. *J. Am. Chem. Soc.* 2014, 136, 4893–4896.

(2) Liu, D.; Xu, X.; Du, Y.; Qin, X.; Zhang, Y.; Ma, C.; Wen, S.; Ren, W.; Goldys, E. M.; Piper, J. A.; et al. Three-Dimensional Controlled Growth of Monodisperse Sub-50 Nm Heterogeneous Nanocrystals. *Nat. Commun.* 2016, 7, 1–8.

(3) Chen, C.; Wang, F.; Wen, S.; Su, Q. P.; Wu, M. C. L.; Liu, Y.; Wang, B.; Li, D.; Shan, X.; Kianinia, M.; et al. Multi-Photon near-Infrared Emission Saturation Nanoscopy Using Upconversion Nanoparticles. *Nat. Commun.* 2018, 9, 4–9.

(4) Wen, S.; Zhou, J.; Zheng, K.; Bednarkiewicz, A.; Liu, X.; Jin, D. Advances in Highly Doped Upconversion Nanoparticles. *Nat. Commun.* 2018, 9, 2415.

REVIEWER COMMENTS

Reviewer #2 (Remarks to the Author):

Summary of key results :

The paper describes rare-earth doped nanorods with varied activator / sensitizer compositions in stratified layers along the axis of a nanorod. The intent of the heterostructure design is to encode information in a spectroscopic fingerprint derived from the varied layers and compositions. Only minor revisions were made to this paper from the most recently reviewed version. This version retains the details of the fabrication and characterization using TEM and a cylindrical beam, demonstrating sub-wavelength localization of the bi-punctate emission. Power dependent color of individual nanorods and on demand generation of white light from nanorods of specific design are demonstrated.

Originality and significance :

The authors have added language to the paper acknowledging the necessity and restriction of these materials for low density labelling applications. The authors demonstrate a fine degree of control in the synthesis of these nanorods and illustrated this with a couple of examples, e.g. power dependent emission color and on demand white light generation from isolated nanorods. While it should be noted that both of these individual effects, in some sense, have been observed previously [4,5], incorporating these effects into the stratified layered nanorods adds some functionality. The paper retains claims of super resolution, but subwavelength localization is probably a more accurate statement. While there is some fine detail in the images of the individual nanorods, likely a result of the narrowing of the point spread function due to the nonlinearity of the rare-earth ion absorption and the beating of the cylindrical excitation profile with the nanorod absorption profile, neither effect seems to be contributing to resolving either one rod from another nor aiding in identifying their spectroscopic signature. The cylindrical point spread function may be important in another context, but I don't see it as critical to either of these points in the experiments shown.

Suggested improvements:

A few more examples demonstrating the independent manipulation of the nanorod degrees of freedom would strengthen the arguments that these are indeed capable of containing a significant amount of information and overall strengthen the paper. A potential application where the nanorods would be ideal carriers, say better than DNA barcodes or other labels would also strengthen the paper. Examining the imaging applications of orientation of the nanorods and the relationship between the cylindrical PSF and the bi-lateral symmetry of the nanorods may also be of interest.

References (including from previous review):

- (1) Zhang, Y.; Zhang, L.; Deng, R.; Tian, J.; Zong, Y.; Jin, D.; Liu, X. Multicolor Barcoding in a Single Upconversion Crystal. *J. Am. Chem. Soc.* 2014, 136, 4893–4896.
- (2) Liu, D.; Xu, X.; Du, Y.; Qin, X.; Zhang, Y.; Ma, C.; Wen, S.; Ren, W.; Goldys, E. M.; Piper, J. A.; et al. Three-Dimensional Controlled Growth of Monodisperse Sub-50 Nm Heterogeneous Nanocrystals. *Nat. Commun.* 2016, 7, 1–8.
- (3) Chen, C.; Wang, F.; Wen, S.; Su, Q. P.; Wu, M. C. L.; Liu, Y.; Wang, B.; Li, D.; Shan, X.; Kianinia, M.; et al. Multi-Photon near-Infrared Emission Saturation Nanoscopy Using Upconversion Nanoparticles. *Nat. Commun.* 2018, 9, 4–9.
- (4) do Carmo, F. F.; do Nascimento, J. P. C.; Façanha, M. X.; Sales, T. O.; Santos, W. Q.; Gouveia-Neto, A. S.; Jacinto, C.; Sombra, A. S. B. White Light Upconversion Emission and Color Tunability in Er³⁺/Tm³⁺/Yb³⁺ Tri-Doped YNbO₄ Phosphor. *J. Lumin.* 2018, 204, 676–684.
- (5) Liao, J.; Jin, D.; Chen, C.; Li, Y.; Zhou, J. Helix Shape Power-Dependent Properties of Single Upconversion Nanoparticles. *J. Phys. Chem. Lett.* 2020, 11, 2883–2890.
- (6) Wen, S.; Zhou, J.; Zheng, K.; Bednarkiewicz, A.; Liu, X.; Jin, D. Advances in Highly Doped Upconversion Nanoparticles. *Nat. Commun.* 2018, 9, 2415.

We were trying to digest what exactly this reviewer has suggested to us in the last two rounds of reviews. By reading both the 3rd and 4th rounds of comments carefully, we realized that his/her comments on the "super resolution" and PSF actually raised from his/her perception that this reviewer believes our nanorods should be used as a new fluorescent tag, and the orientation information inside the bi-punctate nanorods structure can be used for a new mode of super resolution. But the focus of our work is to use the donut beam to resolve the subunits of optical diversities within the rod and selectively excite the subunits using the donut beam scanning approach. This opens new opportunity for sub-nanoparticle optical characterizations.

This reviewer wasn't satisfied by the fact that we have to use low-density of these rods with distance longer than 100 nm between each other, otherwise our current localization accuracy, by telling the twin-spots of the rod, is insufficient to separate different rods. We confirm, our current synthesis of heterogeneous nanorods can be potentially used a new class of fluorescent tags for super resolution imaging and single nanoparticle localization (tracking), but this is not the focus of our current work, plus the size of these rods are relatively large to be used as fluorescent tags.

Also, we insist "super resolution" should be used in this context, as "the spatial resolving power of our single-beam super resolution approach as around 55 nm" is well beyond the diffraction limit, for super resolution characterization of the heterogeneous structure within a single nanorod.

In addition to our early added discussions on Line 152-158 and 245-249. We have further revised our discussions through the text 149-159 (In red) in the main manuscript.

Below are detailed responses (in black) to the new comments from this reviewers (in blue).

Reviewer #2 (Remarks to the Author):

Summary of key results:

The paper describes rare-earth doped nanorods with varied activator / sensitizer compositions in stratified layers along the axis of a nanorod. The intent of the heterostructure design is to encode information in a spectroscopic fingerprint derived from the varied layers and compositions. Only minor revisions were made to this paper from the most recently reviewed version. This version retains the details of the fabrication and characterization using TEM and a cylindrical beam, demonstrating sub-wavelength localization of the bi-punctate emission. Power dependent color of individual nanorods and on demand generation of white light from nanorods of specific design are demonstrated.

Response: We thank the reviewer for his/her summary of our key results.

Originality and significance:

The authors have added language to the paper acknowledging the necessity and restriction of these materials for low density labelling applications. The authors demonstrate a fine degree of control in the synthesis of these nanorods and illustrated this with a couple of examples, e.g. power dependent emission color and on demand white light generation from isolated nanorods. While it

should be noted that both of these individual effects, in some sense, have been observed previously [4,5], incorporating these effects into the stratified layered nanorods adds some functionality.

Response: Our previous work (Ref 4 cited by the reviewer) only shows the power dependent intensity instead of colour, and the core-shell structure of our previously reported material is very different from our current submission.

Power dependent tuning of emission colors using nanorod has never been reported, to our best knowledge.

The on-demand generation of the RGB and white emission from the same single nanorods by selective activation, is significantly different from Ref.5 which is around the synthesis of different kind of microscale materials and their different colours.

The paper retains claims of super resolution, but subwavelength localization is probably a more accurate statement.

Response: We respectfully disagree. Subwavelength localization is a specialized term, not helpful for broad readership.

The resolution achieved in our work, ~ 60 nm, well beyond the diffraction limit, to be called as super resolution characterization of the heterogeneous structure within a single nanorod.

While there is some fine detail in the images of the individual nanorods, likely a result of the narrowing of the point spread function due to the nonlinearity of the rare-earth ion absorption and the beating of the cylindrical excitation profile with the nanorod absorption profile, neither effect seems to be contributing to resolving either one rod from another nor aiding in identifying their spectroscopic signature.

Response: The cylindrical excitation profile contributes to resolving different active sections inside a single nanorod, as well as between different nanorods. Moreover, the cylindrical excitation profile can be used to selectively activate and characterize the spectroscopic signatures of the sub-units of a heterogeneous nanorod (Figures 3 and 4).

The cylindrical point spread function may be important in another context, but I don't see it as critical to either of these points in the experiments shown.

Response: As illustrated in figure 2a, we took the advantage of the non-linear optical responses of upconversion luminescence and its low saturation intensity levels to achieve the small emission point spread function from the relatively large excitation PSF. We think the mechanism has been clearly described in our main text.

Suggested improvements:

A few more examples demonstrating the independent manipulation of the nanorod degrees of freedom would strengthen the arguments that these are indeed capable of containing a significant amount of information and overall strengthen the paper.

Response: This information has been adequately presented through the whole paper.

A potential application where the nanorods would be ideal carriers, say better than DNA barcodes

or other labels would also strengthen the paper. Examining the imaging applications of orientation of the nanorods and the relationship between the cylindrical PSF and the bi-lateral symmetry of the nanorods may also be of interest.

Response: We thank the reviewer for the constructive suggestions for our future works.